# Satellite and Machine Learning Monitoring of Optically Inactive Water Quality Variability in a Tropical River

**Ning Li** [1,2,3,4,5], **Ziyu Ning** [1,2,3,4,5], **Miao Chen** [1,2,3,4,5,*], **Dongming Wu** [1,2,3,4,5], **Chengzhi Hao** [6], **Donghui Zhang** [7,8], **Rui Bai** [9,10], **Huiran Liu** [1,2,3,4,5], **Xin Chen** [1,2,3,4,5], **Wei Li** [1,2,3,4,5], **Wen Zhang** [1,2,3,4,5], **Yicheng Chen** [1,2,3,4,5], **Qinfen Li** [1,2,3,4,5] and **Lifu Zhang** [7,8]

1. Environmental and Plant Protection Institute, Chinese Academy of Tropical Agricultural Sciences, Haikou 571101, China
2. Hainan Danzhou Tropical Agro-Ecosystem National Observation and Research Station, Danzhou 571737, China
3. Key Laboratory of Low-Carbon Green Agriculture in Tropical Region of China, Ministry of Agriculture and Rural Affairs, Haikou 571101, China
4. Hainan Key Laboratory of Tropical Eco-Circular Agriculture, Haikou 571101, China
5. National Agricultural Experimental Station for Agricultural Environment, National Long-Term Experimental Station for Agriculture Green Development, Danzhou 571737, China
6. Hainan Provincial Ecological and Environmental Monitoring Centre, Haikou 571126, China
7. Aerospace Information Research Institute, Chinese Academy of Sciences, Beijing 100094, China
8. Progoo Research Institute, Tianjin Progoo Information Technology Co., Ltd., Tianjin 300380, China
9. College of Resources and Environmental Sciences, China Agricultural University, Beijing 100193, China
10. Hainan Climate Center, Haikou 570203, China
* Correspondence: chenm@catas.cn; Tel.: +86-0898-66969270

**Abstract:** Large-scale monitoring of water quality parameters (WQPs) is one of the most critical issues for protecting and managing water resources. However, monitoring optically inactive WQPs, such as total nitrogen (TN), ammoniacal nitrogen (AN), and total phosphorus (TP) in inland waters, is still challenging. This study constructed retrieval models to explore the spatiotemporal evolution of TN, AN, and TP by Landsat 8 images, water quality sampling, and five machine learning algorithms (support vector regression, SVR; random forest regression, RFR; artificial neural networks, ANN; regression tree, RT; and gradient boosting machine, GBM) in the Nandu River downstream (NRD), a tropical river in China. The results indicated that these models can effectively monitor TN, AN, and TP concentrations at in situ sites. In particular, TN by RFR as well as AN and TP by ANN had better accuracy, in which the $R^2$ value ranged between 0.44 and 0.67, and the RMSE was 0.03–0.33 mg/L in the testing dataset. The spatial distribution of TN, AN, and TP was seasonal in NRD from 2013–2022. TN and AN should be paid more attention to in normal wet seasons of urban and agricultural zones, respectively. TP, however, should be focus on in the normal season of agricultural zones. Temporally, AN decreased significantly in the normal and wet seasons while the others showed little change. These results could provide a large-scale spatial overview of the water quality, find the sensitive areas and periods of water pollution, and assist in identifying and controlling the non-point source pollution in the NRD. This study demonstrated that multispectral remote sensing and machine learning algorithms have great potential for monitoring optically inactive WQPs in tropical large-scale inland rivers.

**Keywords:** machine learning; multispectral remote sensing; optically inactive water quality monitoring; total nitrogen (TN); ammoniacal nitrogen (AN); total phosphorus (TP); tropical river

## 1. Introduction

Freshwater is a vital resource for both the environment and humanity. However, in recent decades, water resources have been threatened by anthropogenic contamination

with the rapid development of the economy and insufficient awareness of environmental protection [1–3]. Due to the indiscriminate discharges of industrial, agricultural, and domestic sewage to water bodies, nutrient concentrations such as nitrogen (N) and phosphorus (P) increase dramatically, leading to non-point source pollution [4,5]. For example, eutrophication, associated algal overgrowth, and declining water quality negatively impact eco-environment services (e.g., drinking water, food supply, and biodiversity) and pose a threat to human health [6–8].

The Nandu River, a tropical river and the largest river on Hainan Island, is one of the primary sources of drinking water, livestock and poultry aquaculture, and agricultural irrigation for Hainan Province, China. It is a typical tropical river basin consisting mainly of agriculture and rural areas, without industry. In recent years, however, the Nandu River has also been facing the problem of eutrophication. Due to the hot and humid climate, the Nandu River basin has had a high multiple cropping index (158.6%) [9] with more nutrient average input per crop season than in temperate regions [10]. Nutrient losses along with the frequent strong rainfall in this area result in the concentrations of total nitrogen (TN), ammoniacal nitrogen (AN), and total phosphorus (TP) exceeding the acceptable range and having great spatial heterogeneity [11–13], as did other tropical rivers around the world [14–16]. Therefore, providing effective strategies to monitor nutrient concentrations of inland water and determining their spatiotemporal dynamics are critical for freshwater resource protection and management [17,18].

In traditional water quality monitoring methods, extensive field monitoring networks are required, including on-site measurements, samplings, and laboratory analysis. Water quality monitoring networks, such as buoys and platforms deployed on some rivers and lakes, can collect data continuously and increase the "temporality" of in situ samplings, but are still spatially limited and costly. Although relatively accurate, most traditional methods are time-consuming, expensive, and limited to point-based data, failing to give a large-scale spatial overview of the water quality [19]. Therefore, researchers and managers aim for a comprehensive, low-cost water quality monitoring method [20].

In recent years, as remote sensing techniques have rapidly developed, satellite remote sensing has become a powerful tool that is widely used for monitoring large-scale variations in water quality [21,22]. Compared to the traditional methods, satellite remote sensing is cost-effective and less time-consuming and has been used as an effective detection tool for the optically active water quality parameters (WQPs), including chlorophyll-a, temperature, turbidity, and suspended solids with high spatiotemporal resolution [23–26]. However, retrieving the N and P concentrations for inland water remains challenging because these WQPs are non-optically active [27]. Fortunately, some scholars have made quantitative retrievals of N and P in bodies of water with empirical or machine learning algorithms [21,28].

Empirical models and machine learning algorithms have been widely used in retrieving WQPs from remote sensing images. The main principle of empirical methods is to use several spectral band combinations as part of polynomial regression to predict the change in water quality [29,30]. They are developed by long-term and large-scale experiments in the fixed water body with high accuracy [31]. Therefore, an empirical model obtained at one water body is not usually applicable for other water bodies, such as tropical rivers having different climates and ambient conditions. With the advances in algorithm development, computing power, sensor performance, and data availability, machine learning algorithms have been widely used for estimating WQPs [27]. Many scholars have used machine learning algorithms and remote sensing data to solve complex regressions such as nonlinear regression, which has relatively obvious advantages in monitoring optically inactive WQPs [32–35]. Popular machine learning algorithms including linear regression, regression tree, random forest, boosted regression tree, support vector regression, and artificial neural networks have been used in water quality monitoring along with various remote sensing data such as Landsat 5/7/8, MODIS, Sentinel 2/3, and airborne systems. Most of these studies focused on TN and TP in reservoirs and lakes with lower flow velocity

in non-tropical regions [29,34,36], few AN, and did not involve in tropical rivers. In order to improve the robustness, these machine learning algorithms must be carefully made generalizable to tropical regions because they have higher temperatures and more frequent precipitation than the other regions. Therefore, a customized model by machine learning algorithms is needed for the monitoring of tropical rivers' N and P concentrations.

The goals of this study are to: (1) develop retrieval models of WQPs to monitor the concentrations of TN, AN, and TP in the Nandu River based on Landsat 8 images, machine learning algorithms, and water quality samplings; (2) evaluate the accuracy of these models and determine the best model for a long-term analysis from 2013 to 2022; and (3) explore the spatiotemporal distribution and patterns of N and P concentrations and identify the sensitive area and period in the tropical river.

## 2. Materials and Methods

### 2.1. Study Area

The Nandu River downstream (NRD, 19°41′49″–20°4′56″N, 109°58′38″–110°26′7″E) is in the tropical area of northern Hainan Island, China (Figure 1). The waterway, which runs about 110.7 km from the Shankou national cross-section to the river mouth, flows into the Qiongzhou strait at the Haikou City Xinbu delta and occupies a total drainage area of ~2290 km$^2$. It is a primary river system that flows through agricultural, rural, and urban areas, mostly hilly tableland and coastal deltas. This area has a tropical monsoon marine climate with an annual mean temperature of ~24.6 °C, long summers, and no winters. The annual precipitation is ~1880.6 mm (1992–2021), most of which occurs in tropical cyclones or typhoons from July to October, and the spatiotemporal distribution is highly uneven. The annual average runoff is ~6.92 billion m$^3$. The dry season (January–March), wet season (June–October), and normal season (April, May, November, December) recur throughout the year. Therefore, the area has a better agricultural foundation and a high level of intensive cultivation and breeding, with an industrial advantage in the areas of planting, freshwater aquatic, livestock, and poultry. To reveal the spatial variations of the nutrients in the different environments, this study divided the NRD into three subregions (Figure 1). More generally, the NRD was divided into an agriculture and rural mixed zone (A), an agriculture zone (B), and an urban zone (C) based on location, shape, geomorphology, and ecological function.

### 2.2. Data

#### 2.2.1. Water Quality Sampling

Surface water quality data include the national cross-sections and the in situ site samplings. Five national cross-sections were obtained from the Environmental Knowledges Service System (http://envi.ckcest.cn/environment/data_Integration/data_Integration.jsp accessed on 18 April 2022) at the NRD from January 2021 to March 2022. The data are hourly (4 h) including the concentrations of TN, AN, and TP. The value at 12:00 was used due to being near the Landsat 8 scene center time in the NRD. Based on the five national cross-sections and field surveys, nine in situ water quality sites were established with representativeness, easy sampling, stability, and reliability as supplementary monitoring in the NRD. In situ water samples were collected once a month from the water surface (~20 cm) in the time frame of March 2021 to September 2021, stored in acid-washed Niskin bottles under ice conditions, and delivered to the laboratory on time. Then, WQPs were analyzed, including the concentrations of TN, AN, and TP. In order to test the TN concentration, the solutions were digested with potassium persulfate, and then a UV spectrophotometer was used to characterize the sample at a wavelength of 210 nm. The spectrophotometry running at the wavelength of 620 nm using Nessler's reagent was used to test the AN concentration. The TP concentration was tested by spectrophotometry at 700 nm using ammonium molybdate. The environmental quality standards for surface water (GB 3838-2002) in China (Table S1) were used to evaluation water quality. There are five levels,

including classes I, II, III, IV, and V according to the standard limit values of WQPs from low to high.

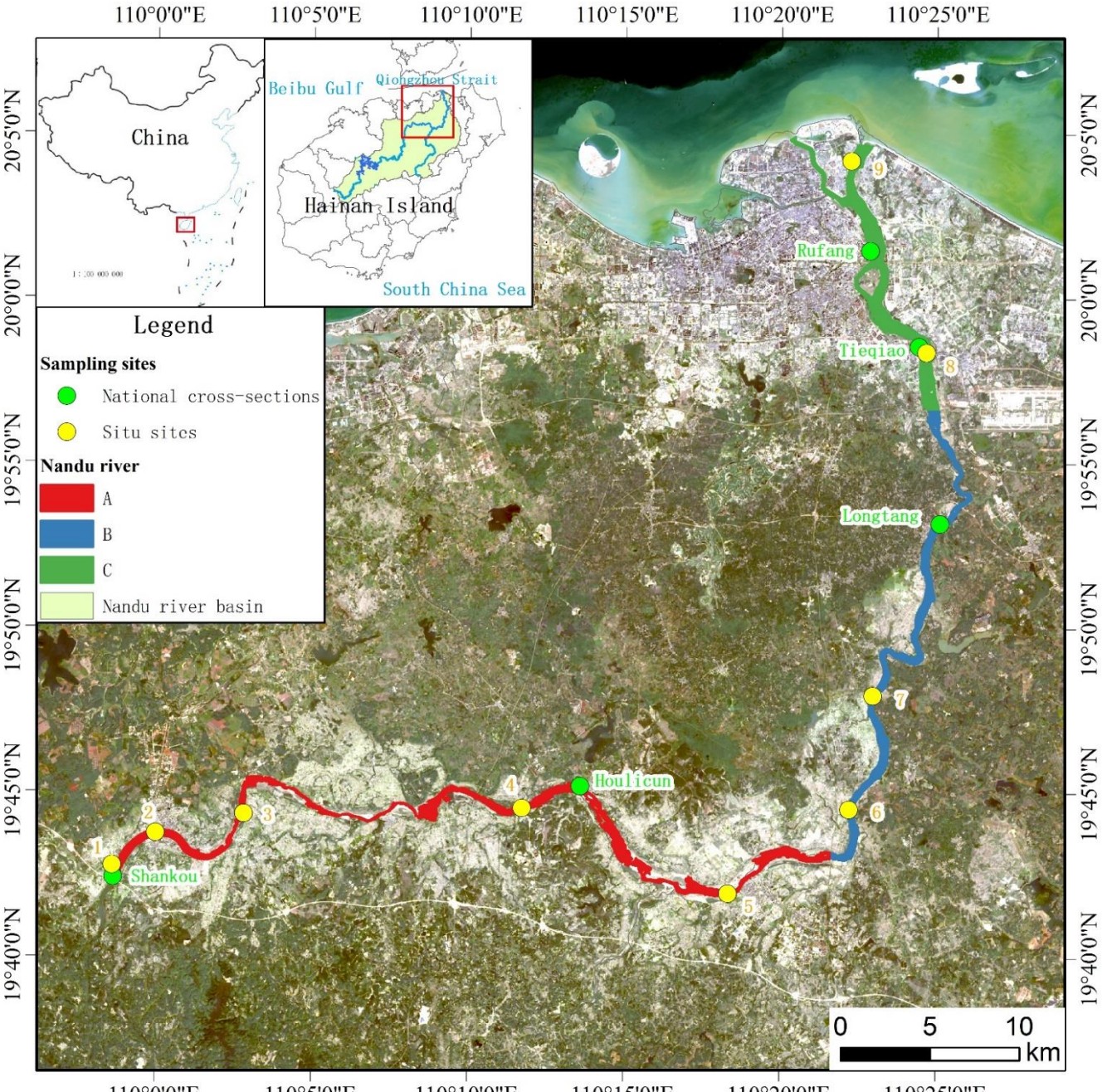

**Figure 1.** Location of the Nandu River downstream and distribution of sampling sites. The map of China is produced under the supervised by the Ministry of Natural Resources of China with the map content approval number: GS(2016)1554.

### 2.2.2. Landsat 8 Satellite Images

This study used Level-2 surface reflectance products in Landsat 8 operational land imager collection 2, which can be freely downloaded from the website of the United States Geological Survey (USGS) (http://earthexplorer.usgs.gov accessed on 19 May 2022). These products are radiometrically and geometrically corrected. Landsat 8 was successfully launched by NASA on 11 February 2013 and had a 16-day revisit cycle. This study used bands 1–7 of Landsat 8 with a spatial resolution of 30 m and a spectrum range from visible

to short-infrared. In this study, 25 Landsat 8 images were collected with little clouds over the NRD from April 2013 to April 2022 at row 46 and path 124 (Table 1) to explore the spatiotemporal evolution of TN, AN, and TP in the NRD. There were totals of 3, 11, and 11 images at the dry, normal, and wet seasons, respectively.

**Table 1.** Temporal distribution of Landsat 8 images used in this study.

| Season | Dry | | | Normal | | | | Wet | | | Normal | | Total |
|---|---|---|---|---|---|---|---|---|---|---|---|---|---|
| Year | January | February | March | April | May | June | July | August | September | October | November | December | |
| 2013 | \ | \ | \ | 1 | 1 | 0 | 0 | 0 | 0 | 1 | 0 | 1 | 4 |
| 2014 | 0 | 0 | 0 | 0 | 0 | 0 | 0 | 0 | 0 | 0 | 0 | 0 | 0 |
| 2015 | 0 | 0 | 0 | 0 | 1 | 2 | 0 | 0 | 0 | 0 | 1 | 0 | 4 |
| 2016 | 0 | 0 | 1 | 0 | 0 | 0 | 0 | 0 | 0 | 0 | 0 | 1 | 2 |
| 2017 | 0 | 0 | 0 | 0 | 0 | 0 | 0 | 1 | 0 | 0 | 0 | 0 | 1 |
| 2018 | 0 | 0 | 0 | 0 | 0 | 0 | 1 | 0 | 0 | 0 | 0 | 0 | 1 |
| 2019 | 0 | 0 | 0 | 0 | 1 | 2 | 1 | 0 | 1 | 0 | 0 | 0 | 5 |
| 2020 | 1 | 0 | 0 | 0 | 1 | 1 | 0 | 1 | 0 | 0 | 0 | 0 | 4 |
| 2021 | 1 | 0 | 0 | 0 | 1 | 0 | 0 | 0 | 0 | 0 | 0 | 1 | 3 |
| 2022 | 0 | 0 | 0 | 1 | \ | \ | \ | \ | \ | \ | \ | \ | 1 |
| Total | 2 | 0 | 1 | 2 | 5 | 5 | 2 | 2 | 1 | 1 | 1 | 3 | 25 |

### 2.2.3. Dataset for Modeling

There were 67 water samples collected, including national cross-sections ($N = 39$) and in situ sites ($N = 28$) with quasi-synchronous Landsat 8 images ranging from January 2021 to March 2022. The concentrations of TN, AN, and TP for all pixel–sample matchups were in the ranges of 0.30–2.97 mg/L, 0.03–0.65 mg/L, and 0.01–0.30 mg/L, respectively, covering the typical range for the NRD. The dataset, used to construct the retrieval models for TN, AN, and TP, was combined with the surface water quality data of 67 samples (output data) and the spectral values of seven bands of Landsat 8 images at the corresponding water samples (input data). The dataset was randomly divided into two groups: the training dataset and testing dataset. The training dataset contained 47 (AN) and 42 (TN and TP) samples, which accounted for 70% of the data and were used to train the parameters of retrieval models. The rest of dataset as the testing dataset contained 20 (AN) and 18 (TN and TP) samples that were used to test the performance of retrieval models.

### 2.3. Methods

#### 2.3.1. Support Vector Regression (SVR)

SVR is a classical machine learning approach due to its robust ability to capture nonlinear trends. The basic idea of SVR [37] is that the original data points can be mapped from the input space into a feature with higher or even infinite spatial dimensions, where an optimal separating hyperplane with the minimum distance to all data points is established. It has been widely used in water environment prediction [38,39]. In this study, SVR was performed by the radial basis function of package "e1071" [40] in the R language.

#### 2.3.2. Random Forest Regression (RFR)

RFR is a modern nonparametric technique for nonlinear multiple regression [41] and has been one of the best machine learning algorithms in recent years. As an ensemble algorithm, RFR constructs a large set of decision trees (typically hundreds or several thousand) through integrated learning, using one decision tree as the basic unit and each decision tree as a predictor. The composition of the forest is relatively simple, consisting mainly of independent and disconnected decision trees. The result is predicted from the average results of all decision trees with high accuracy and generalization performance. In addition, RFR is efficient on large datasets, can evaluate the significance of each input

feature, and can effectively avoid overfitting [42]. The "randomForest" [43] package of R language was used to implement this regression technique.

### 2.3.3. Artificial Neural Networks (ANNs)

ANNs are similar to the mediation of the human brain, with a nerve cell model as a numerical analysis unit. The back-propagation neural network algorithm [44] used in this study is the most commonly used ANN in simulating the nonlinear relationship between variables. Thus, ANNs play a vital role in artificial intelligence. There are two processes in the primary learning of this algorithm, i.e., a forward computation process and an error back-propagation process. ANNs consist of three main layers: the input layer, hidden layer, and output layer. Neurons in different layers are connected by the corresponding weights, which can be updated in the error back-propagation process by back-propagating the output error in some form layer-by-layer through the hidden layer to the input layer, assigning the error to individual neural units of each neuron in each layer. The details of the ANN can be found in previous studies [21,36,45]. The "neuralnet" package [46] in R language was used to implement the ANN using the "neuralnet" function.

### 2.3.4. Regression Tree (RT)

RT analysis has been widely used for remote sensing-based environmental monitoring [47]. Generally speaking, the RT model can be considered as a binary splitting process in which the feature space is recursively stratified into subdivisions. Each node has two possible values, i.e., "Yes" or "No". The basis of stratification is to minimize the deviation from the mean of the response variable. This rule-based regression model consists of four parts, i.e., root nodes, internal nodes, branch nodes, and terminal nodes. The prediction accuracy of the model is highly related to the total number of nodes. The redundant growth of the model leads to a higher possibility of overfitting, which can reduce the prediction accuracy of the test set. Therefore, the necessary pruning of the tree is required. RT analysis can also be applied in nonlinear regression modeling to interpret the relationships between the independent and dependent variables in the form of rulesets. The "rpart" package [48] in the R language was used in this study.

### 2.3.5. Gradient Boosting Machine (GBM)

The GBM is another popular machine learning algorithm that has the advantages of high accuracy, a fast training process, short prediction time, and a small memory footprint in various applications [49]. Similar to RFR, the GBM [50] also consists of an ensemble of decision trees, but the sequence of trees is created, and each tree in the sequence focuses on the previous tree's prediction residuals. The innovation of the GBM is its use of a nonparametric approach to estimate the basis function and using gradient descent to approximate the solution in function space. It does not depend on the pre-processing of variables, it can handle missing data, and it can solve overfitting problems. The "gbm" [51] package in the R language was used in this study.

### *2.4. Accuracy Assessment*

The accuracy of these WQP retrieval models was evaluated by the coefficient of determination ($R^2$), the root-mean-square error (RMSE), and the mean absolute percentage error (MAPE). The training and testing of these models, the statistical analysis of model parameters, the calculation of correlation coefficients, and the error analysis were mainly implemented by the R 4.2.0 language with the "raster", "dplyr", "e1071", "randomForest", "neuralnet", "rpart", and "gbm" packages. ArcGIS 10.2.2 and R with the "ggplot2" package were used to map the nutrient concentrations in the NRD spatially and temporally.

## 3. Results

### 3.1. Model Training and Testing

The WQP retrieval models for TN, AN, and TP were constructed by SVR, RFR, ANN, RT, and GBM algorithms through training and testing. ANN, RFR, and SVR showed better prediction results than RT and GBM for TN, AN, and TP (Figure 2, Table 2). Since the regression results were close to classification and narrowed the prediction range, the RT and GBM algorithms were unsuitable for the prediction of these WQPs. In addition, the $R^2$ of RT and GBM were below 0.55, and the average was 0.29 in the training and testing datasets (Figure 2(a4,b4,c4,a5,b5,c5), and Table 2).

**Table 2.** Comparison of different machine learning algorithms for the training and testing datasets.

| WQP | Dataset | Model | Slope | Intercept | $R^2$ | $p$ | RMSE (mg/L) | MAPE (%) |
|---|---|---|---|---|---|---|---|---|
| TN | Training N = 42 | SVR | 0.24 | 0.70 | 0.50 | <0.01 | 0.44 | 24.39 |
| | | RFR | 0.42 | 0.62 | 0.75 | <0.01 | 0.34 | 27.50 |
| | | ANN | 0.97 | 0.03 | 0.97 | <0.01 | 0.09 | 7.70 |
| | | RT | 0.21 | 0.82 | 0.21 | <0.01 | 0.48 | 38.69 |
| | | GBM | 0.06 | 0.99 | 0.31 | <0.01 | 0.51 | 40.20 |
| | Testing N = 18 | SVR | 0.18 | 0.73 | 0.20 | 0.06 | 0.41 | 35.05 |
| | | RFR | 0.30 | 0.70 | 0.49 | <0.01 | 0.33 | 33.53 |
| | | ANN | 1.04 | 0.01 | 0.45 | <0.01 | 0.48 | 33.11 |
| | | RT | 0.25 | 0.71 | 0.17 | 0.08 | 0.40 | 38.32 |
| | | GBM | 0.07 | 0.97 | 0.25 | 0.04 | 0.40 | 43.59 |
| AN | Training N = 47 | SVR | 0.21 | 0.17 | 0.31 | <0.01 | 0.14 | 202.56 |
| | | RFR | 0.44 | 0.13 | 0.66 | <0.01 | 0.10 | 153.12 |
| | | ANN | 0.99 | 0.00 | 0.99 | <0.01 | 0.01 | 6.09 |
| | | RT | 0.37 | 0.14 | 0.37 | <0.01 | 0.13 | 169.79 |
| | | GBM | 0.22 | 0.17 | 0.55 | <0.01 | 0.13 | 195.37 |
| | Testing N = 20 | SVR | 0.06 | 0.19 | 0.07 | 0.25 | 0.16 | 305.05 |
| | | RFR | 0.16 | 0.18 | 0.24 | 0.03 | 0.15 | 284.84 |
| | | ANN | 0.96 | −0.05 | 0.44 | <0.01 | 0.19 | 318.07 |
| | | RT | 0.14 | 0.18 | 0.17 | 0.08 | 0.15 | 273.77 |
| | | GBM | 0.05 | 0.20 | 0.13 | 0.11 | 0.16 | 312.73 |
| TP | Training N = 42 | SVR | 0.58 | 0.04 | 0.66 | <0.01 | 0.03 | 59.51 |
| | | RFR | 0.60 | 0.03 | 0.86 | <0.01 | 0.02 | 30.01 |
| | | ANN | 0.69 | 0.02 | 0.69 | <0.01 | 0.03 | 46.23 |
| | | RT | 0.23 | 0.05 | 0.23 | <0.01 | 0.04 | 49.24 |
| | | GBM | 0.06 | 0.06 | 0.46 | <0.01 | 0.04 | 59.89 |
| | Testing N = 18 | SVR | 0.26 | 0.05 | 0.59 | <0.01 | 0.04 | 52.53 |
| | | RFR | 0.14 | 0.05 | 0.21 | 0.06 | 0.04 | 48.23 |
| | | ANN | 0.60 | 0.03 | 0.67 | <0.01 | 0.03 | 46.44 |
| | | RT | 0.21 | 0.05 | 0.24 | 0.04 | 0.04 | 54.84 |
| | | GBM | 0.06 | 0.06 | 0.42 | <0.01 | 0.05 | 64.46 |

The ANN had the best prediction for TN in the training dataset, with the $R^2$ above 0.95 and the RMSE around 0.09 mg/L (Figure 2(a3)). However, according to the values of the $R^2$ and RMSE, the ANN performed worse than RFR ($R^2 = 0.49$ and RMSE = 0.33 mg/L) in the testing dataset (Figure 2(a2)). RFR also had a better prediction in the training dataset ($R^2 = 0.75$ and RMSE = 0.34 mg/L). Thus, RFR is the best model for TN. For the prediction of AN, the algorithms are ranked as ANN, RFR, and SVR from the best to the worst in the training and testing datasets. Furthermore, the ANN had the highest $R^2$ values, which were close to 1 and 0.44 for the training and testing datasets, respectively, and the RMSE was less than 0.20 mg/L (Figure 2(b3)). RFR had the highest $R^2$ (0.86) and the lowest RMSE (0.02 mg/L) in the training dataset for TP, but its $R^2$ and RMSE values were the lowest in the testing dataset (Figure 2(c2)). The ANN was the best in the testing dataset ($R^2 = 0.67$ and RMSE = 0.03 mg/L) and similar to RFR in the training dataset (Figure 2(c3)). Therefore, ANN is the best method for TP.

In general, these models constructed by Landsat 8 and machine learning algorithms can consistently monitor TN (by RFR), AN, and TP (both by ANN) at the national cross-section and in situ site samplings. The spread of predicted and measured WQPs was close to the 1:1 line. In addition, these WQPs had significant linear correlations for the testing dataset (Table 2), where the $R^2$ ranged between 0.44 and 0.67, and the RMSE was 0.03–0.33 mg/L. The results indicated that these selected models could quantitatively monitoring TN, AN, and TP in the study area.

### 3.2. Spatiotemporal Variability of N and P Concentrations

TN, AN, and TP concentrations in the NRD were monitored for the whole region (All) and three subregions (A, B, and C) in the dry, normal, and wet seasons from 2013 to 2022 using the selected models. Temporal change analysis was not conducted in the dry season due to the small amount of Landsat 8 images ($N = 3$).

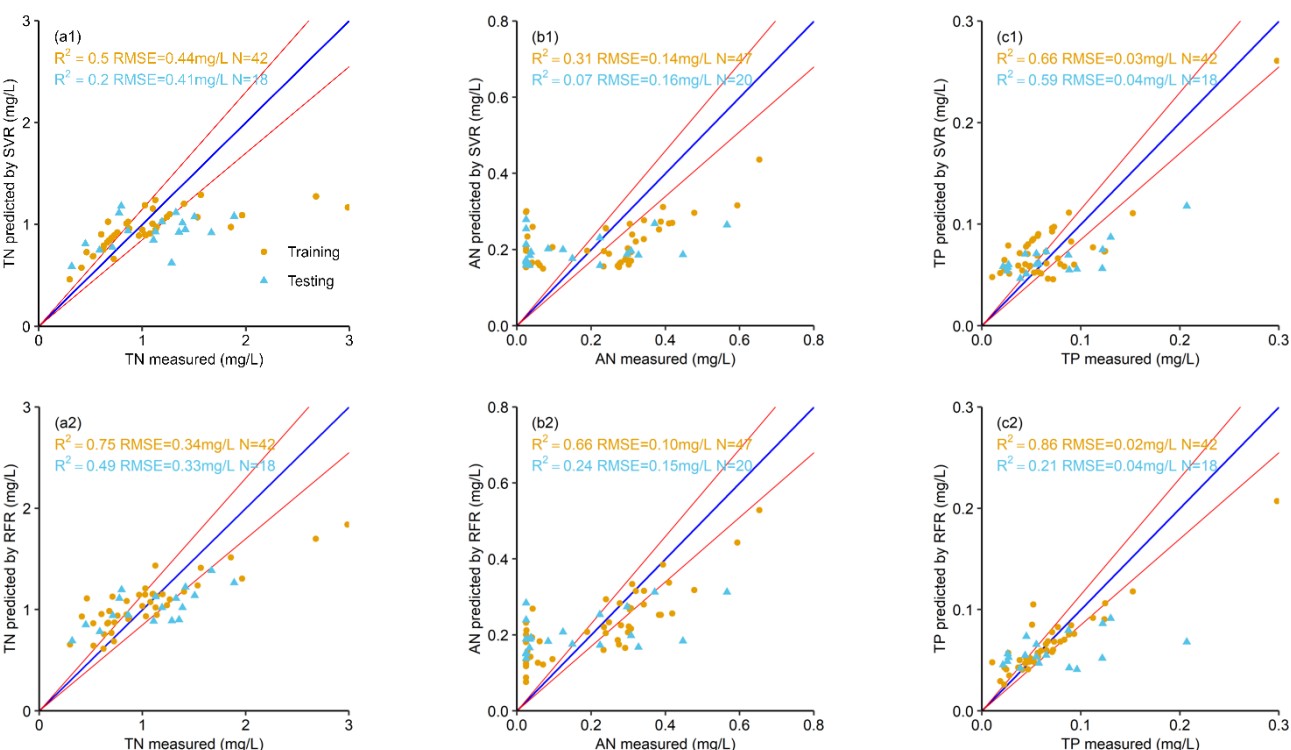

**Figure 2.** *Cont.*

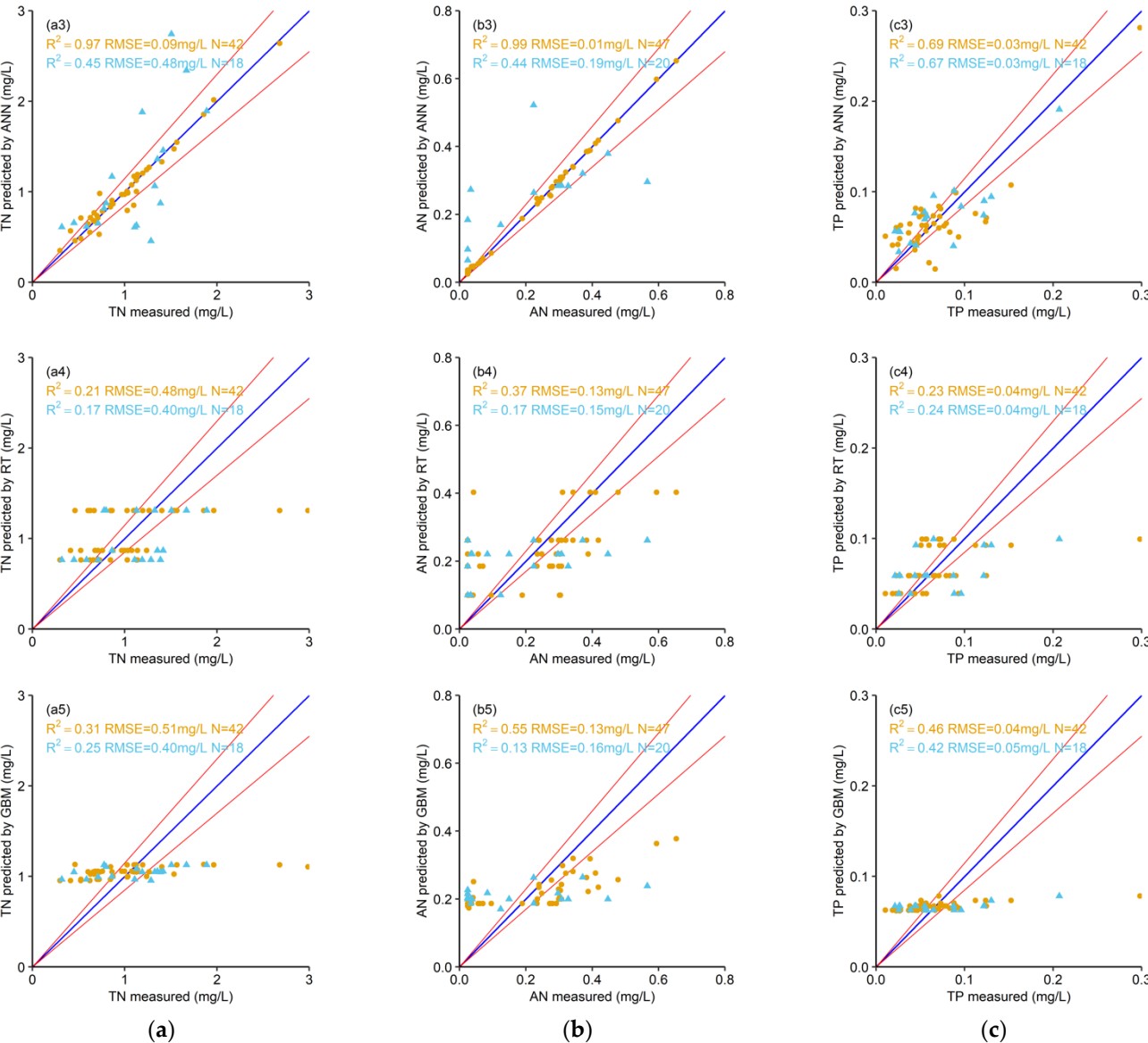

**Figure 2.** Relationships between in situ measured and Landsat 8 predicted TN (**a**), AN (**b**), and TP (**c**) for pixel–sample matchups. (**1**–**5**) indicate different machine learning algorithms: SVR, RFR, ANN, RT, and GBM, respectively. The orange text and circles represent the training results, and the blue text and triangles represent the testing results. The blue line is 1:1, and the red line is 15% variation.

The TN concentration, ranging from 0.69 to 1.72 mg/L (Figures 3 and S1) in class III to V water quality, was low in the dry season (mean $1.09 \pm 0.11$ mg/L) with the lowest concentration in region A, and high in the wet season (mean $1.27 \pm 0.20$ mg/L) with the highest concentration in region C. In all seasons, the maximum value of TN always occurred in region C, followed by region B. In addition, the spatial distribution of the TN concentration varied enormously in the normal and wet seasons but was more consistent in the dry season. Temporally, the TN had no significant change in the wet and normal seasons (Table S2, Figures 4 and S1). The highest value ($1.44 \pm 0.17$ mg/L) of TN was monitored in region C in 2016, and the lowest value ($1.06 \pm 0.11$ mg/L) was monitored in region A in 2019 in the NRD's normal season.

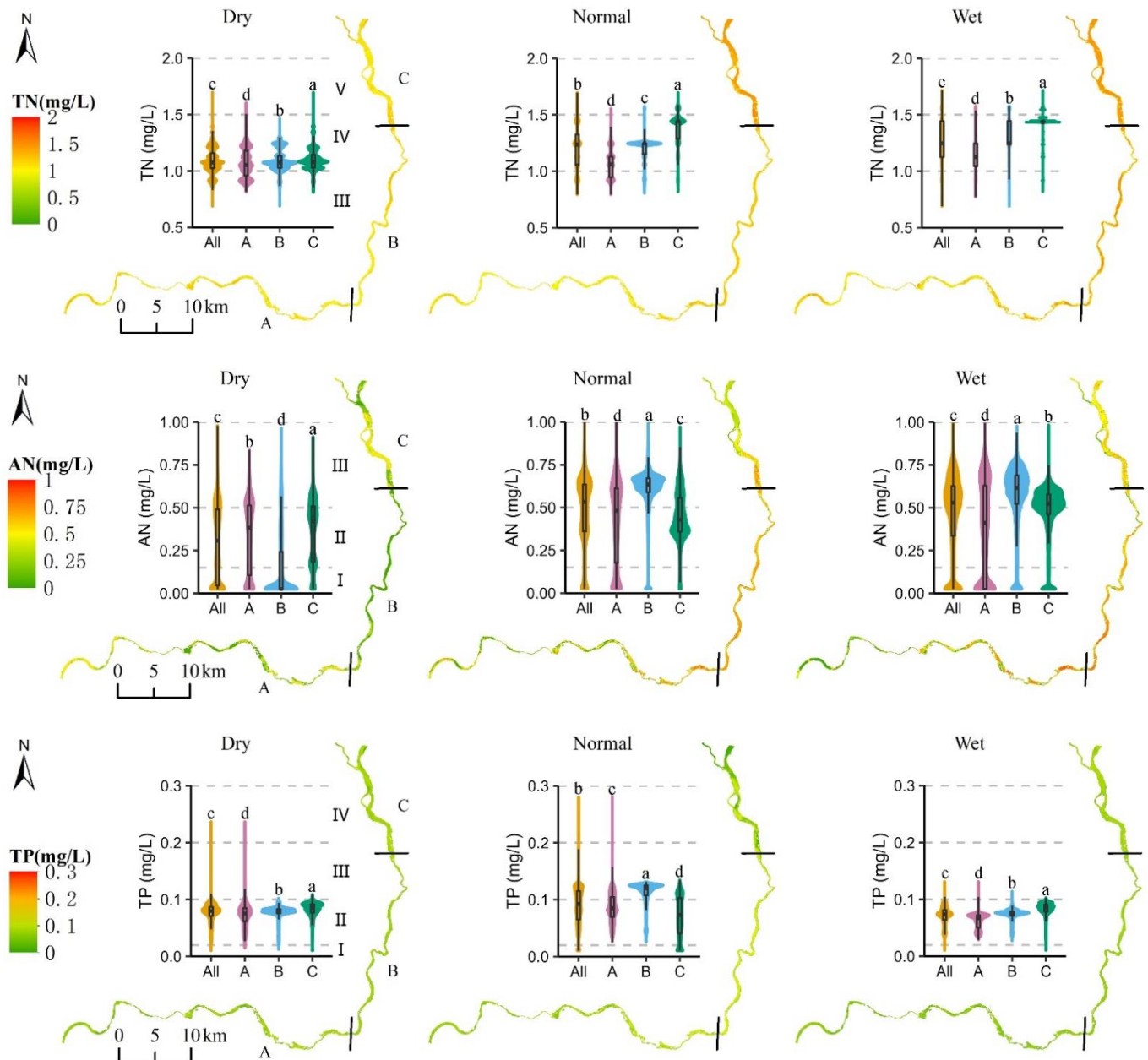

**Figure 3.** Spatial distributions of mean Landsat 8-monitored TN, AN, and TP concentrations in the whole region of the NRD (All) and three subregions (A, B, and C) in the dry, normal, and wet seasons from 2013 to 2022. Different lowercase letters indicate significant differences at the level of 0.05. I, II, III, IV, and V are the quality standards for surface water from low to high.

The AN concentration, ranging between 0.03 and 1.21 mg/L (Figures 3 and S2) in class I to III water quality, was low in the dry season (mean 0.29 ± 0.21 mg/L) and high in the normal season (mean 0.48 ± 0.21 mg/L); the lowest and the highest concentrations both occurred in region B. The maximum concentration of AN in the dry season was monitored in region C, followed by region A; and in the normal and wet seasons, it was monitored in region B, followed by region C. Moreover, the spatial distribution of AN varied strongly in all seasons, especially in the dry season. Temporally, the AN concentration decreased significantly in the normal and wet seasons, especially in the normal season with a rate of 0.02 mg/a (Table S3, Figures 5 and S2). The highest (0.71 ± 0.21 mg/L) and lowest (0.08 ± 0.14 mg/L) concentrations of AN were in region B's wet seasons of 2015 and 2020, respectively.

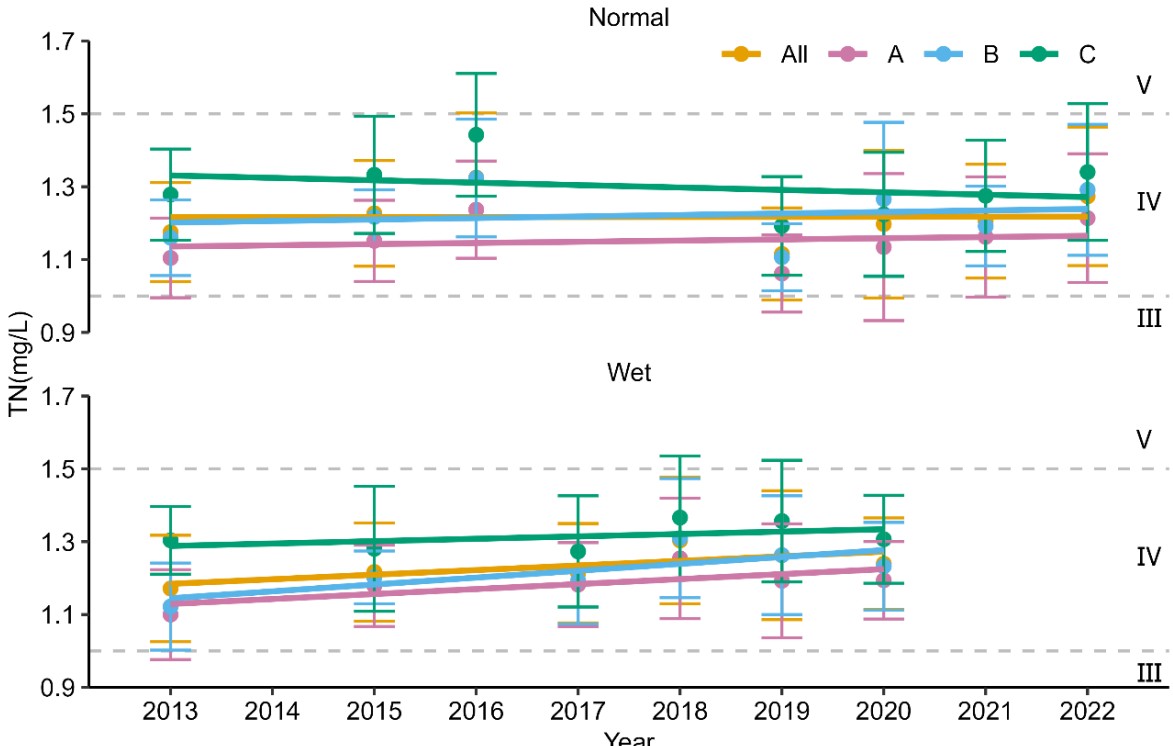

**Figure 4.** Time series of mean Landsat 8-monitored TN of the NRD in the whole region (All) and three subregions (A, B, and C) in the normal and wet seasons from 2013 to 2022. III, IV, and V are the quality standards for surface water from low to high.

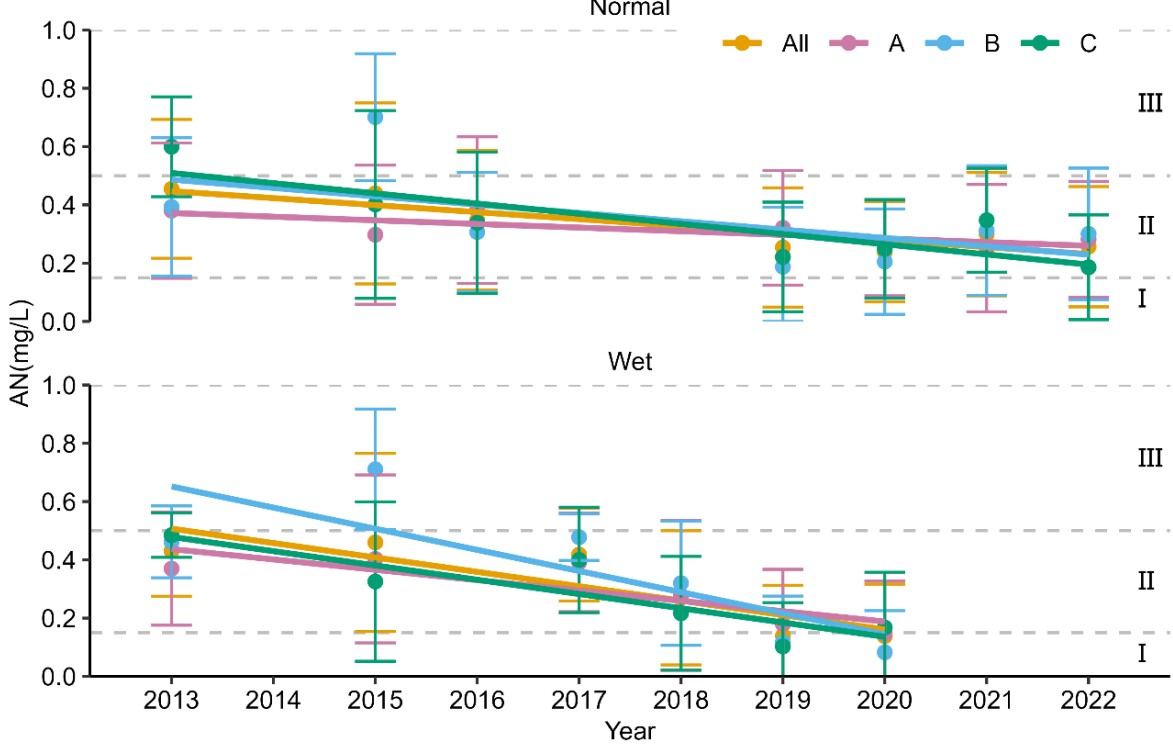

**Figure 5.** Time series of mean Landsat 8-monitored AN of the NRD for the whole region (All) and three subregions (A, B, and C) in dry, normal, and wet seasons from 2013 to 2022. I, II, and III are the quality standards for surface water from low to high.

The TP concentration, ranging from 0.01 to 0.28 mg/L (Figures 3 and S3) in class I to IV water quality, was low in the wet season (mean 0.07 ± 0.02 mg/L) with the lowest concentration in region A, and high in the normal season (mean 0.09 ± 0.03 mg/L) with the highest concentration in region B. In the dry and wet seasons, the maximum concentration of TP occurred in region C, followed by region B; and in the normal season, the maximum concentration of TP occurred in region B, followed by region A. In addition, the spatial distribution of TP varied strongly in the dry and normal seasons but was more consistent in the wet season. Temporally, the decrease rate of TP passed the significance test only in the normal season in region A but was close to 0 (Table S4, Figures 6 and S3). The highest value (0.10 ± 0.01 mg/L) of TP was observed in the wet season of 2013 in region C, and the lowest value (0.06 ± 0.01 mg/L) was observed in the normal season of 2015 in region C.

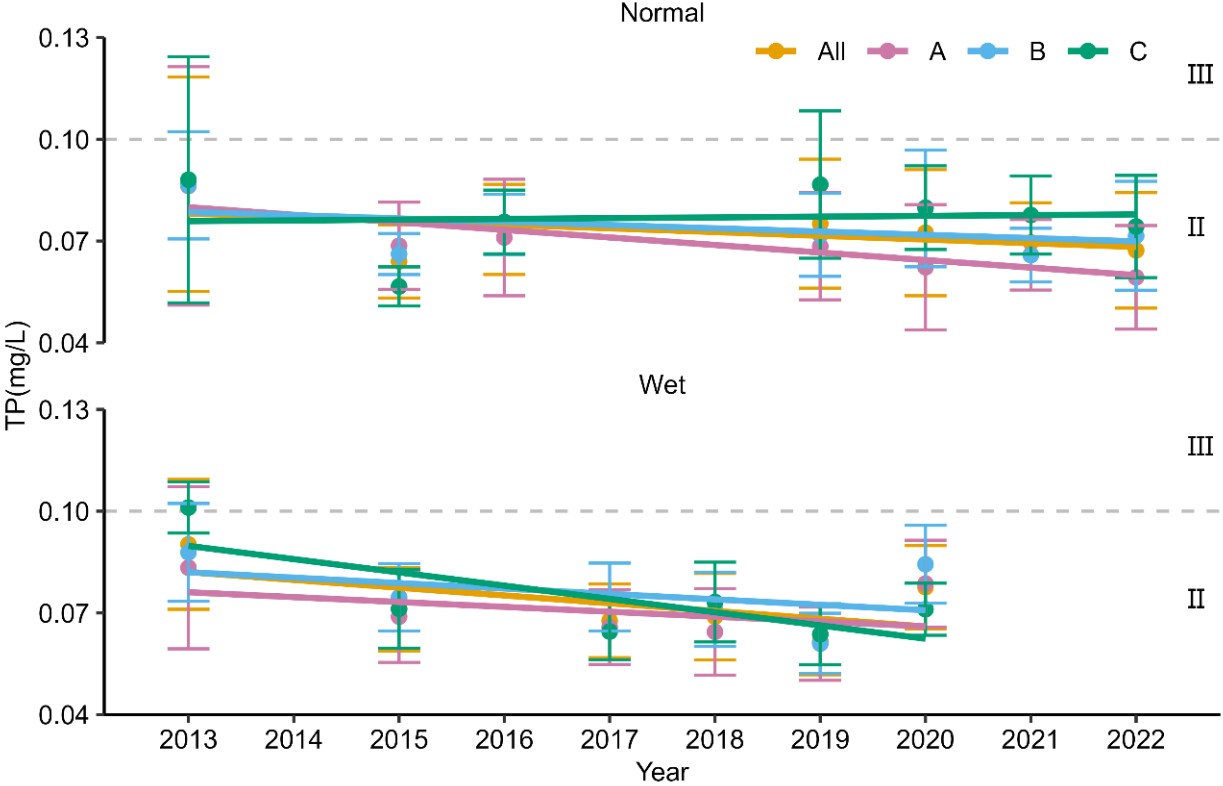

**Figure 6.** Time series of mean Landsat 8-monitored TP of the NRD for the whole region (All) and three subregions (A, B, and C) in dry, normal, and wet seasons from 2013 to 2022. II and III are the quality standards for surface water from low to high.

## 4. Discussion

### 4.1. Models Comparison and Future Optimization

The results show that the concentrations of TN, AN, and TP, especially TP, can be accurately monitored faster, easier, and more low-cost by machine learning algorithms and Landsat 8 data in the tropical river. In these models, the underlying hydrologic and environmental processes don't need to be explicitly described in a mathematical form [52]. Therefore, these models can be used without many variables. In other words, these constructed models only need a sufficient amount of surface water quality monitoring data and remote sensing data from the same period.

Using field measurements, sampling, and laboratory analysis to monitor the concentration of WQPs is expensive and time-consuming. Currently, the remote sensing retrieval models have not completely replaced experimental analysis. However, the pixels of Landsat 8 images are equivalent to setting a sampling point in a 30 × 30 m² water area. Thus, Landsat 8 has a much higher spatial sampling density than conventional sampling. Higher

resolution can better indicate large-scale changes in nutrient concentrations [38]. It can further help to optimize the location of water samples in watersheds and maximize the effectiveness of the data collection strategy. In the long term, satellite observations highly reduce uncertainty due to the sampling intervals and the uneven spatial distribution of sampling. It is of great significance to help fully interpret the changes and rules of river nutrient concentrations and the contribution of various non-point source pollution.

As a complex biogeochemical system, the aquatic ecosystem contains numerous chemical, physical, and biological components that undergo a large range of integrated transformation processes. It is challenging to find the associations between WQPs and bands of remote sensing in bodies of water. The existing regression equations had high accuracy when fitted in fixed waters. However, when they were extended and applied to other waters, the results were relatively coarse. This study also compared the monitoring performance of the selected models with previous regression equations [29,30,53] for TN and TP concentrations in the NRD. Since there are few regression equations existing for AN, it is not compared in the study. Figure 7, Tables S5 and S6 indicate that the previous regression equations applicable to specific regions are difficult to apply to monitoring TN and TP in the NRD. Most empirical methods establish statistical relationships between nutrient concentrations and remotely sense reflectance on the basis of linear, exponential, or logarithmic regressions [36,54,55] without strict theoretical foundations. Furthermore, the regression fits for single and multiband combinations were susceptible to changes in the specified bands. Thus, they were poorly generalized and did not apply to other conditions or locations [56].

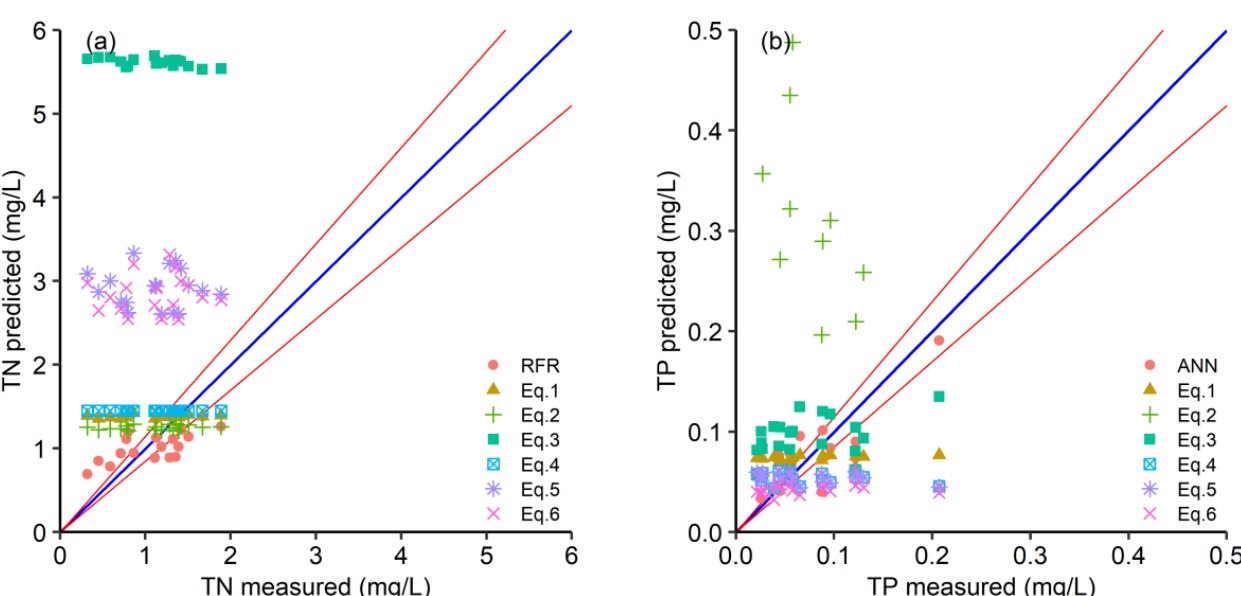

**Figure 7.** Comparison of prediction performance between machine learning algorithms and previous regression equations for TN (**a**) and TP (**b**) in the NRD.

While excellent model performance was demonstrated in this study, some additional aspects need to be further investigated in future studies. More advanced deep learning and artificial intelligence algorithms with large training and testing datasets from various rivers should be used to avoid overfitting and improve the monitoring accuracy and robustness of the retrieval model. Considering the uncertainties of the model parameters, structure, and input data, monitoring results' uncertainty should be quantified to facilitate data interpretation [39].

*4.2. Water Quality Evaluation*

This study area is located in the Haikou, Dingan, and Chengmai regions of Hainan Province, China. From the Hainan statistical yearbook [9], this area has a resident pop-

ulation of ~3.67 million (75.66% in urban), a cultivated land area of ~$10.29 \times 10^4$ ha, a freshwater aquatic area of ~7742 ha, chemical fertilizer consumption of ~$26.46 \times 10^4$ t, numbers of livestock and poultry raised in the whole year of ~162.17 and ~5645.26 million heads, and little industry in 2020, respectively. The area is a typical tropical agricultural town non-point source pollution area, and the pollution source is mainly the discharges of agriculture and domestic sewage [57].

The results indicated the seasonality of TN, AN, and TP distribution in the NRD. TN and AN concentrations were high in the wet and normal seasons and low in the dry season. This result was similar to previous studies on the Yiluo River [58], the Ru River [18], and the Yangtze River [30]. The wet and normal seasons almost overlap with the rainy season in this area. Increased rainfall could result in N being exported into the water body through runoff. Rainfall often imports a lot of N from agricultural and domestic sources [59,60]. Another reason may be N deposition. Chen et al. [61] found that there was more atmospheric N deposition in summer than in other seasons. In the urban region, wastewater is probably the largest nutrient source [62,63], and this area has structurally unsound sewers and relatively simple sewage treatment processes. Sewers often leak at low flows and may surcharge and overflow during the inflow and infiltration in wet weather, which is related to the age of infrastructure, the pipe materials, and the dimensions of pipes [64,65]. Thus, the NRD showed a gradually decreasing distribution of N concentration from the estuary to the upstream with the inflow of other tributaries.

However, TP concentration was high in the normal season and low in the wet season. This result was similar to previous studies on the Pearl River [66] and the Fuyang River [67]. In contrast to N, there are more forms of P existing in water bodies, especially dissolved P, which can be adsorbed on mineral surfaces [68]. Thus, P is easily present in sediments by binding to aluminum oxides, interiors of iron, and calcium oxide [69]. Replenishing many clean water bodies in the NRD has diluted the TP of surface water in the wet season. The replenishment decreases during the normal and dry seasons, aquatic plant residue decomposes, and sediments release nutrients into surface water, increasing the TP. In addition, Berthold et al. [70] reported that for coastal waters, P deposition is a considerable nutrient flux. These may be the reasons for the order of TP in these regions, i.e., C > B > A, in the wet and dry seasons.

Temporally, AN had a significant decreasing trend in the normal and wet seasons. The reason for this is the significant investments in wastewater discharge standards and pollution control strategies that were made to address the water pollution crisis and promote eco-environment restoration [5].

### 4.3. Limitations

Ideally, the more Landsat 8 images are used, the more accurate the retrieval model will be, and the N and P concentrations' spatial distribution and temporal trends will be more clearly identified in the NRD. However, this study only obtained a limited number of Landsat 8 images due to the coarse temporal (16 days) and spatial (30 m) resolutions and the heavy cloud cover in tropical areas. These disadvantages limit the application of satellite remote sensing for inland water research in tropics [35].

Furthermore, satellite observations are almost instantaneous, and therefore the high temporal variability of nutrient concentrations is an important consideration. The results of 24 h continuous TN, AN, and TP monitoring that grouped different subregions and hydrological seasons in the NRD are shown in Figure 8. Nutrient concentrations varied slightly throughout the day except on a few individual dates, and the scene center time of Landsat 8 does not overlap the time in which the extremum value occurs. Therefore, the satellite monitoring results are representative of the daily mean values. Further studies should focus on achieving high-temporal-resolution (e.g., hourly) monitoring of nutrient concentrations in a river by using geostationary satellite images [71,72] or assimilating remote sensing and water quality models [73–75].

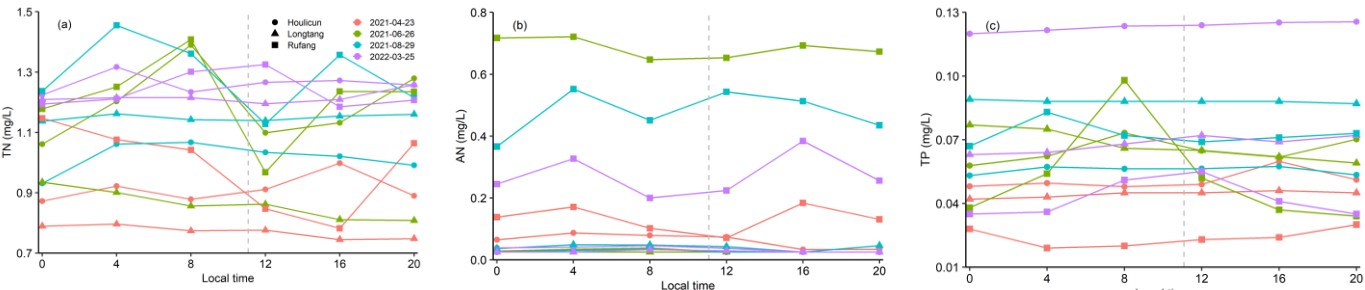

**Figure 8.** Diurnal cycle of TN (**a**), AN (**b**), and TP (**c**) measured at national cross-sections. Vertical dashed lines at 11:05 represent the Landsat 8 scene center time in the NRD.

In addition, hyperspectral remote sensing is a feasible solution to improve the accuracy of the WQP retrieval model. It can obtain abundant information about inland water bodies, making it possible to accurately identify nutrient concentrations due to high spatial and spectral resolution [76]. It has been applied to inland water quality monitoring, such as TN, TP, AN, cyanobacterial blooms, chlorophyll-a, total suspended matter, and colored dissolved organic matter [33,35,77–80]. However, portable, proximal, and unmanned aerial vehicle-borne hyperspectral data can only observe limited areas. The entire area of the NRD is difficult to explore because it has about 110.7 km of waterways. Furthermore, trade-offs among spectral, spatial, and radiometric resolutions limit the spatial resolution of the spaceborne hyperspectral images to medium or coarse resolutions [81,82]. Moreover, few long-term series continuous data of hyperspectral remote sensing are now available for temporal variation studies. Therefore, multispectral remote sensing is a better choice for studying the spatiotemporal variation of WQPs in large-scale inland areas.

## 5. Conclusions

Machine learning algorithms for monitoring optically inactive WQPs were constructed in this study by Landsat 8 images and ground monitoring data. The TN concentration can be effectively predicted by RFR, and AN and TP by ANN. These retrieval models had much higher spatial resolutions than conventional field sampling and were more accurate than previous regression equations for TN and TP in the NRD, which is a large-scale tropical inland river. They can be used to interpret the changes and rules of TN, AN, and TP in the NRD. The spatiotemporal evolution of TN, AN, and TP in the NRD was explored in the dry, normal, and wet seasons at the pixel level from 2013 to 2022. The concentrations of N and P showed significant seasonal changes. In the normal and wet seasons, AN had a significant decreasing trend, while TN and TP had almost no significant change. Considering the supervision and management of non-point source pollution in the NRD, TN and AN should be paid more attention to in the normal and wet seasons of urban and agricultural zones, respectively; TP, however, should be focused on in the normal season of agriculture zones. The results can give a large-scale spatial overview of the water quality, identify the sensitive areas and periods with water pollution, provide an important basis for the supervision and management of non-point source pollution in the whole region, and be applied to other similar tropical rivers. This study demonstrated that multispectral remote sensing, currently, is a better choice for studying the spatiotemporal variation of optically inactive WQPs in large-scale and long-term tropical inland waters. Further studies can use more advanced deep learning and artificial intelligence algorithms with hyperspectral remote sensing to improve the robustness and accuracy of these retrieval models in different water bodies. In the future, the retrieval model could be extended for larger tropical regions and more WQPs, assist in identifying the pollution source, and provide insight into nutrient dynamics and transfers from inland to ocean waters.

**Supplementary Materials:** The following supporting information can be downloaded at: https://www.mdpi.com/article/10.3390/rs14215466/s1, Table S1: classification standards of water quality levels in GB 3838–2002; Table S2: statistical information of TN for all regions and seasons; Table S3: statistical information of AN for all regions and seasons; Table S4: statistical information of TP for all regions and seasons; Table S5: statistical information of RFR for TN compare with previous regression equations in the testing dataset; Table S6: statistical information of ANN for TP compare with previous regression equations in the testing dataset; Figure S1: the spatial distribution of TN in NRD; Figure S2: the spatial distribution of AN in NRD; Figure S3: the spatial distribution of TP in NRD.

**Author Contributions:** Conceptualization, N.L. and M.C.; methodology, N.L., D.Z. and M.C.; software, N.L. and Z.N.; validation, N.L. and M.C.; formal analysis, N.L.; investigation, N.L., H.L., X.C., W.L., W.Z. and Y.C.; resources, N.L., C.H. and R.B.; data curation, N.L., H.L., W.L., W.Z. and Y.C.; writing—original draft preparation, N.L., Z.N. and M.C.; writing—review and editing, N.L., Z.N., D.Z., M.C., D.W. and R.B.; visualization, N.L. and Z.N.; supervision, M.C., Q.L. and L.Z.; project administration, N.L. and M.C.; funding acquisition, N.L., M.C. and Q.L. All authors have read and agreed to the published version of the manuscript.

**Funding:** This research was funded by the Major Science and Technology Plan of Hainan Province, China (ZDKJ2021008), the Key Research and Development Project of Hannan Province, China (ZDYF2022XDNY210), and the Central Public-interest Scientific Institution Basal Research Fund, China (1630042022022).

**Data Availability Statement:** The data used in this study are available from the corresponding website.

**Acknowledgments:** The authors would like to acknowledge the data provided by the Environmental Knowledges Service System, the United States Geological Survey (USGS), and the anonymous reviewers for their constructive comments and suggestions to improve this manuscript.

**Conflicts of Interest:** The authors declare no conflict of interest.

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
