# Peer review of "Satellite and Machine Learning Monitoring of Optically Inactive Water Quality Variability in a Tropical River"

_remotesensing, doi:10.3390/rs14215466_

Round 1

Reviewer 1 Report

Title: Satellite and Machine Learning Monitoring Optically Inactive Water Quality Variability in A Tropical River

Abstract: Large-scale monitoring of water quality parameters (WQP) is one of the most critical issues for protecting and managing water resources. However, monitoring optically inactive WQP, such as total nitrogen (TN), ammoniacal nitrogen (AN), and total phosphorus (TP) in inland waters, is still challenging. This study constructed retrieval models to explore the spatiotemporal evolution of TN, AN, and TP by Landsat 8 images, water quality sampling, and five machine learning algorithms (support vector regression, SVR; random forest regression, RFR; artificial neural networks, ANN; regression tree, RT; and gradient boosting machine, GBM) in Nandu River downstream (NRD), a tropical river, China. The result indicated that these models can effectively monitor TN, AN, and TP at in-situ sites. Especially, TN by RFR, AN and TP by ANN had better accuracy, in which the R2 ranged between 0.44 and 0.67, and the RMSE was 0.03 - 0.33 mg/L in the testing dataset. The spatial distribution of TN, AN and TP was seasonal in NRD from 2013 - 2022. TN and AN should pay more attention in normal wet seasons of urban and agricultural zone, respectively. TP, however, should focus on the normal season of the agriculture zone. Temporally, AN decreased significantly in the normal and wet seasons and the others showed little obvious change trend. These results can provide a significant scientific reference for TN, AN, and TP monitoring and non-point source pollution control in NRD. This study demonstrated that multispectral remote sensing and machine learning algorithms have great potential for monitoring optically inactive WQP in tropical large-scale inland rivers.

Comments:

1.       The authors used Landsat 8 to predict the water quality parameters using a few machine learning algorithms, however, the methods used are not very new and been used for years for water quality prediction. This needs to be explained a little

2.       The methods have been explained with enough detail and the accuracy measures are properly described

3.       Figure 2 looks confusing as it is not mentioned whether these results are for training or testing or both

4.       The ANN model for TN and ANN is clearly overfitting as seen in the training and testing R2. What can be the reason for those issues? Is it due to the sample size or the complexity of the model?     

Author Response

Thanks for you and your comments, which greatly improve this manuscript. We carefully revised the manuscript in terms of these constructive comments; the point-to-point responses to these comments can be found below. Hopefully it can meet the publication’s requirements of Remote Sensing.

Reviewer 2 Report

Add more info about the limit/gap of similar studys in this field

Section: Materials and Methods - add info about the dates of Landsat images – I do not have access to Table S1

Water sampler are collected between March 2021 to September 2021 (r. 144) but on r. 163-166 used images from Apr. 2013 to Apr. 2022 – please explain.

Please rewrite r. 138-140 or subsection Water quality sampling and provide info about the water samples from 2013-2020

Results or Methods: add some info about the sub-sections (A, B, C) evaluated in figures 3, 4, 5, 6 and how was selected or add a references to figure 1 (info is present in legend)

Fig. 1 increase the size of yellow and green points to be more visible

Why do not use the same number of samples for training and testing for An comparing with TN and TP (Table 1) – please explain

It is possible to provide some maps with spatial distribution of TN, AN and TP values on the entire river (using ANN model  or ....) ?

What is the practical implication of your study? It is possible to identify some pollution point sources?

And in subsection “Limitations and future directions” some sentences about possible utility of your methods – you discus about possible hourly monitoring of water quality.

Author Response

Thank you for your affirmation and your constructive suggestions for this manuscript. According to those suggestions, we have corrected the mistakes and marked them in red. Our point-by-point replies are listed below.

Round 2

Reviewer 1 Report

The authors responded to the question very well. I agree that the manuscript bepublished in remote sensing. All the best